# Cerebrospinal Fluid *α*-Synuclein Species in Cognitive and Movements Disorders

**DOI:** 10.3390/brainsci11010119

**Published:** 2021-01-17

**Authors:** Vasilios C. Constantinides, Nour K. Majbour, George P. Paraskevas, Ilham Abdi, Bared Safieh-Garabedian, Leonidas Stefanis, Omar M. El-Agnaf, Elisabeth Kapaki

**Affiliations:** 1Neurochemistry and Biomarkers Unit, 1st Department of Neurology, National and Kapodistrian University of Athens, 11528 Athens, Greece; geoprskvs44@gmail.com (G.P.P.); ekapaki@med.uoa.gr (E.K.); 2Ward of Cognitive and movement Disorders, 1st Department of Neurology, National and Kapodistrian University of Athens, 11528 Athens, Greece; lstefanis@med.uoa.gr; 3Neurological Disorders Research Centre, Qatar Biomedical Research Institute (QBRI), Hamad Bin Khalifa University (HBKU), Qatar Foundation, Doha 34110, Qatar; nmajbour@hbku.edu.qa (N.K.M.); iyahya@hbku.edu.qa (I.A.); oelagnaf@hbku.edu.qa (O.M.E.-A.); 4College of Medicine, Member of QU Health, Qatar University, Doha 2713, Qatar; bsafieh@qu.edu.qa

**Keywords:** α-synuclein, cerebrospinal fluid, biomarkers, tau proteins, beta amyloid, parkinsonism, dementia: neurodegenerative disorders

## Abstract

Total CSF α-synuclein (t-α-syn), phosphorylated α-syn (pS129-α-syn) and α-syn oligomers (o-α-syn) have been studied as candidate biomarkers for synucleinopathies, with suboptimal specificity and sensitivity in the differentiation from healthy controls. Studies of α-syn species in patients with other underlying pathologies are lacking. The aim of this study was to investigate possible alterations in CSF α-syn species in a cohort of patients with diverse underlying pathologies. A total of 135 patients were included, comprising Parkinson’s disease (PD; *n* = 13), multiple system atrophy (MSA; *n* = 9), progressive supranuclear palsy (PSP; *n* = 13), corticobasal degeneration (CBD; *n* = 9), Alzheimer’s disease (AD; *n* = 51), frontotemporal degeneration (FTD; *n* = 26) and vascular dementia patients (VD; *n* = 14). PD patients exhibited higher pS129-α-syn/α-syn ratios compared to FTD (*p* = 0.045), after exclusion of samples with CSF blood contamination. When comparing movement disorders (i.e., MSA vs. PD vs. PSP vs. CBD), MSA patients had lower *α-*syn levels compared to CBD (*p* = 0.024). Patients with a synucleinopathy (PD and MSA) exhibited lower t-*α-*syn levels (*p* = 0.002; cut-off value: ≤865 pg/mL; sensitivity: 95%, specificity: 69%) and higher *pS129-*/t-*α-*syn ratios (*p* = 0.020; cut-off value: ≥0.122; sensitivity: 71%, specificity: 77%) compared to patients with tauopathies (PSP and CBD). There are no significant α-syn species alterations in non-synucleinopathies.

## 1. Introduction

Diagnostic accuracy of cognitive and movement disorders based on established clinical diagnostic criteria is suboptimal, even among experts [1,2,3,4]. This is due to the clinical variability of most neurodegenerative diseases, which, not uncommonly, manifest with atypical phenotypes.

Biomarkers that provide in vivo information of the underlying pathology in patients with cognitive and movement disorders are needed in order to improve diagnostic accuracy. Cerebrospinal fluid (CSF) biomarkers have been extensively studied to this end, due to their relatively low cost (especially compared to PET imaging), safety of lumbar puncture and the proximity of CSF to brain parenchyma.

CSF biomarkers have already been incorporated in the NIA/AA and IWG-2 diagnostic criteria for Alzheimer’s disease (AD) [5,6]. These include amyloid beta with 42 amino acids (Aβ_42_), a marker of amyloid pathology (plaque formation); total tau protein (τ_T_), a non-specific marker of neurodegeneration; phosphorylated tau protein at threonine 181 (τ_P-181_), a marker of tau pathology (tangle formation). AD is characterized by a decrease in Aβ_42_ and an increase in τ_T_ and τ_P-181_, which results in elevated biomarker indices such as τ_T_/Aβ_42_ and τ_P-181_/τ_T_ ratios.

Despite the success of CSF and imaging biomarkers in AD, clinically relevant biomarkers in synucleinopathies, such as Parkinson’s disease (PD), Parkinson’s disease dementia (PDD), dementia with Lewy bodies (DLB) and multiple system atrophy (MSA) are still lacking.

CSF total *α*-synuclein (t-*α*-syn) has been studied as a surrogate biomarker of synucleinopathies [7,8,9,10,11]. Most studies agree that CSF t-*α*-syn is decreased by 10–20% in PD compared to healthy subjects and other neurodegenerative disorders. However, there is considerable overlap among neurodegenerative diseases, resulting in suboptimal diagnostic accuracy. Results on other synucleinopathies such as DLB and MSA have produced mixed results [7,8,9].

Substantial evidence supports the notion that oligomeric (o-) and phosphorylated Ser129 (pS129-) *α*-syn species underlie and drive the neurodegenerative process in synucleinopathies [12,13,14] and various CSF *α*-synuclein species have been tested as biomarkers of synucleinopathies, with mixed results. These studies point towards an increase in CSF pS129-*α*-syn and/or *o-α*-syn in PD and other synucleinopathies [8,10,15,16,17,18].

Most studies on CSF α-syn species have focused exclusively on synucleinopathies. Few studies have compared *α*-syn species among other proteinopathies, such as tauopathies [17,19]. Limited data from these studies suggest that AD and PSP patients may have higher CSF t-*α*-syn or unchanged *o-α*-syn levels compared to patients with PD, PDD or DLB.

Intriguingly, it has been suggested that t-*α*-syn is elevated in AD compared to healthy controls, and that t-*α*-syn may correlate with τ_T_ in AD. Moreover, a significant increase in t-*α*-syn has been reported in Jacob–Creutzfeldt disease, as is the case for τ_T_. These findings may imply that t-*α*-syn could represent a non-specific marker of neurodegeneration, as is the case for τ_T_. CSF *α*-syn species have not been systematically studied in neurodegenerative disorders of different underlying pathologies.

The primary aim of this study was to investigate the diagnostic accuracy of three different species of *α* -syn (t-, o- and pS129-α-syn) in a cohort of patients with cognitive and/or movement disorders of diverse underlying pathologies (synucleinopathies and tauopathies), AD (a mixed tauopathy and amyloidosis) and vascular dementia. A secondary objective was to explore any possible correlations between *α*-syn species and established CSF AD biomarkers (τ_Τ_, τ_P181_, Aβ_42_). We also investigated if any correlations with clinical parameters and α-syn species are present.

## 2. Materials and Methods

### 2.1. Patients

The medical files of all patients, with available CSF AD biomarkers (τ_T_, τ_P-181_, Aβ_42_), examined at the Wards of Cognitive and Movement disorders of the First Department of Neurology of the Medical School of the National and Kapodistrian University of Athens at Eginition Hopsital, between 2011 and 2017, were retrospectively reviewed. Patients fulfilling established clinical diagnostic criteria for clinically established PD [20], and probable MSA [21], AD [5], Richardson’s syndrome of progressive supranuclear palsy (PSP) [22], corticobasal degeneration (CBD) [23], frontotemporal dementia (FTD) [24] and vascular dementia (VD) [25] were included. All patients underwent an extensive standardized work-up, including brain MRI, to exclude other diagnoses.

Patients with a typical CSF AD profile (defined as elevated τ_T_
*and* τ_P-181_
*and* decreased Aβ_42_) [26,27] and a different clinical diagnosis were excluded from analysis. For this purpose, the cut-off points of the Unit of Neurochemistry and Biomarkers were used (i.e., τ_Τ_ ≤ 376 pg/mL; τ_P-181_ ≤ 57 pg/mL; Aβ_42_ ≥ 682 pg/mL). In the absence of pathological data, these patients were excluded from analysis, because they may represent AD patients with atypical clinical presentation (e.g., behavioral-frontal syndrome, corticobasal syndrome) or patients with dual pathologies.

### 2.2. Cerebrospinal Fluid Markers

Different species of CSF *α-*syn were measured including t-, o- and pS129-*α*-syn following previously published sandwich-based enzyme-linked immunosorbent assays (ELISA) [28]. For all assays, samples were run in duplicate and measured in a blinded fashion. Diagnostic groups were randomized over the plates. A series of internal controls were also run to normalize for inter-run variation. Specified calibrators were used to generate an 8-point standard curve to which a 4-parameter logistic (4PL) curve of all plates was fitted and used to quantify unknown concentrations using GraphPad Prism software.

AD CSF biomarkers (τ_T_, τ_P-181_, Aβ_42_) were measured in duplicate by double sandwich, ELISA by commercially available kits (“Innotest^®^ hTau antigen”, “β-amyloid1–42” and “phospho-tau181”, respectively, Fujirebio Europe, Gent, Belgium) according to manufacturer instructions. A protocol controlling for pre-analytical and analytical variability was implemented, with regard to classical CSF biomarker analyses [29]. This protocol is described in detail in previous studies [30,31].

### 2.3. Statistical Analysis

Shapiro–Wilk’s and Levene’s tests were used to test for normality of distribution and homogeneity of variance, respectively. Demographic and clinical data among study groups were compared by *x^2^* and analysis of variance (ANOVA) as appropriate. Neuropsychological and movement disorder test scores were compared by analysis of covariance (ANCOVA), with sex, age and disease duration as covariates or independent samples Kruskal-Wallis test as appropriate.

CSF biomarkers did not have normal distributions. Log-transformation restored normality of distributions for τ_T_, Aβ_42_, τ_P-181_, pS129-α-syn and O-α-syn/*α*-syn ratios. These variables were analyzed by ANCOVA, with sex, age and disease duration as covariates. Post-hoc analysis with Bonferroni correction for multiple comparisons was applied. All other CSF biomarkers were compared by Kruskal–Wallis, as appropriate.

In addition to τ_T_, τ_P-181_, Aβ_42_, *α*-syn, *pS129-α*-syn and *o-α-*syn, the following ratios were calculated: τ_T_/Aβ_42_, τ_P-181_/τ_T_, *pS129-α-*syn/*a-*syn and *o-α-*syn/*α*-syn.

The following analyses were performed: (a) an initial analysis included all study groups; (b) patients with movement disorders (i.e., PD vs. MSA vs. PSP vs. CBD); (c) patients with dementia (i.e., AD vs. FTD vs. VD); (d) patients based on presumed underlying proteinopathy (i.e., synucleinopathy vs. tauopathy).

These analyses were repeated for *α-*syn species, after exclusion of patients with CSF blood contamination. This was defined as RBC > 50 cells/mm^3^, as this cut-off value was determined to minimize the effect of CSF blood contamination in α-syn levels [32]. Non-parametric tests were used for these analyses due to the small number of patients per study group, as appropriate.

In cases of statistical significance, Receiver Operating Characteristics (ROC) Curve analysis was performed to define the cut-off point that provided optimal combined sensitivity and specificity.

Spearman’s rank correlation coefficient was applied to investigate possible correlations between CSF biomarkers and clinical data, as well as among CSF biomarkers within different study groups. To correct for multiple comparisons, significance level was set at *p* < 0.001 for correlation analyses.

Analyses were performed by IBM SPSS Statistics^®^ version 23.0.0.0 (SPSS Inc., Chicago, IL, USA 2013). Graphs were designed using GraphPad Prism^®^, version 5.03 (GraphPad Software Inc., La Jolla, CA, USA 2009).

## 3. Results

A total of 135 patients were included, comprising PD (*n* = 13), MSA (*n* = 9), PSP (*n* = 13), CBD (*n* = 9), AD (*n* = 51), FTD (*n* = 26) and VD patients (*n* = 14). The study groups differed in age (*p* = 0.001), disease duration (*p* = 0,010), MMSE score (*p* < 0.0001), Clox 2 test (*p* = 0.002) and 5-word delayed recall test (*p* = 0.003). Post-hoc analysis revealed that VD patients were older than PD (*p* = 0.011) and FTD patients (*p* = 0.014). AD patients performed poorer in neuropsychological tests compared to other study groups (Table 1).

The initial analysis included all study groups. CSF *α*-syn species did not differ significantly among groups in this analysis (Table 2, Figure 1). However, after exclusion of patients with CSF blood contamination, study groups differed in the pS129-α-syn/α-syn ratio (*p* = 0.039). Post-hoc analysis revealed that PD patients had significantly higher ratios compared to FTD (*p* = 0.045) (Table 2, Appendix A).

There were statistically significant differences in τ_Τ_, τ_P-181_, Aβ_42_, τ_T_/Aβ_42_ ratio and τ_P-181_/τ_T_ ratio (*p* < 0.0001). Post-hoc analysis revealed that these differences were attributable to AD patients, who had: (a) higher τ_T_ levels compared to MSA (*p* = 0.003), PSP (*p* < 0.0001), PD (*p* < 0.0001), FTD (*p* = 0.001) and VD (*p* = 0.012); (b) higher τ_P-181_ compared to MSA (*p* = 0.0020), PSP (*p* < 0.0001), PD (*p* = 0.036) and FTD (*p* < 0.0001); (c) lower Aβ_42_ levels compared to CBD (*p* = 0.039), PSP (0.018), PD (*p* = 0.027) and FTD (*p* < 0.0001); (d) higher τ_T_/Aβ_42_ ratio compared to PD and PSP (*p* < 0.0001) and FTD (*p* = 0.001); e) higher τ_P-181_/τ_T_ compared to PD (*p* = 0.016) (Table 2).

The second analysis included movement disorders (i.e., PD vs. MSA vs. PSP vs. CBD). There was no statistically significant difference regarding *α*-synuclein species, when all patients were included (Table 2). After excluding CSF samples with blood contamination, *α-*syn differed significantly among study groups (*p* = 0.016). This was attributed to lower *α-*syn levels in MSA compared to CBD (*p* = 0.024) (Appendix A, Table 2).

Statistically significant differences were found in τ_Τ_ and the τ_T_/Aβ_42_ ratio. Post-hoc analysis revealed that these differences were due to higher τ_T_ levels in CBD compared to PSP (*p* = 0.017) and higher τ_T_/Aβ_42_ ratios in CBD compared to PD (*p* = 0.002) and PSP (*p* = 0.046) (Table 2). The third analysis compared dementias (AD vs. FTD vs. VD). CSF *α-*syn species did not differ among groups. AD patients had greater τ_T_, τ_P-181_, τ_T_/Aβ_42_ ratio and τ_P-181_/τ_T_ ratio and lower Aβ_42_ levels compared to VD and FTD (*p* < 0.0001 for all measurements, *p* = 0.022 for τ_P-181_/τ_T_ ratio) (Table 2).

The final analysis compared patients based on presumed underlying proteinopathy (i.e., synucleinopathies vs. tauopathies). CSF t-*α*-syn was significantly lower in presumed synucleinopathies compared to presumed tauopathies (*p* = 0.034). ROC curve analysis revealed that a value of t-*α-*syn ≤ 865 pg/mL provided a sensitivity of 86% and a specificity of 59% for a synucleinopathy diagnosis (Table 3, Appendix A, Figure 2).

The cohort of patients with no CSF blood contamination included 94 patients (PD:9; MSA:8; PSP:6; CBD:7; AD:37; FTD:16; VD:11). In this cohort, synucleinopathies exhibited significantly lower t-*α-*syn levels (*p* = 0.002) and higher *pS129-*/t-*α-*syn ratios (*p* = 0.020). ROC curve analysis indicated that t-*α-*syn ≤865 pg/mL produced a 95% sensitivity and 69% specificity for a synucleinopathy diagnosis. A *pS129-α-*syn/*α-*syn ratio value ≥0.122 provided 71% sensitivity and 77% specificity for a synucleinopathy diagnosis (Table 3, Appendix A).

In the AD group, τ_T_ correlated significantly with τ_P-181_ (r = 0.855, *p* < 0.001. In the PSP group there was significant correlation between τ_T_ and τ_P-181_ (r = 0.944, *p* < 0.001), τ_T_ and Aβ_42_ (r = 0.867, *p* < 0.001) and τ_P-181_ and Aβ_42_ (r = 0.951, *p* < 0.001). There were no correlations between CSF biomarkers in any of the other study groups. CSF biomarkers did not correlate with any of the clinical or demographic characteristics in any of the study groups (Appendix A).

## 4. Discussion

The in vivo recognition of the underlying pathology in neurodegenerative disorders is the initial step towards the development of disease-modifying, protein-targeted treatments. To this end, CSF Aβ_42_, τ_Τ_ and τ_P-181_ are established AD biomarkers, which can predict an underlying Alzheimer’s disease pathology with high diagnostic accuracy. Biomarkers of other proteinopathies (e.g., synucleinopathies, TDP-43 proteinopathies) are presently lacking.

CSF *α*-syn species have been studied to this end, including t-*α-*syn, and, more importantly, *o-α-*syn and *pS129-α-*syn. Most studies in the field have focused on differences of *α-*syn levels and species among synucleinopathies, or between synucleinopathies and AD or PSP, reporting lower t-α-syn and higher *pS129-α-*syn and *o-α-*syn values in synucleinopathies [7,8,10,15,16,28,33]. *α* -Syn oligomers were increased in plasma of PD and brain extracts of DLB patients [34,35]. A comprehensive evaluation of different *α*-syn species in a variety of movement and cognitive disorders is presently lacking.

The present study examined CSF *α*-syn species (t-, *o-* and *pS129-α*-syn) in a variety of neurodegenerative disorders of presumed underlying proteinopathy. These included synucleinopathies (PD, MSA), tauopathies (PSP, CBD) and mixed tauopathy with amyloidosis (AD). Moreover, we further included patients with FTD (either 3R-tauopathy or TDP-43-proteinopathy) and VD, as paradigms of diseases with neuronal death due to neurodegeneration and ischemia, respectively.

In our cohort, CSF t-*α*-syn was numerically lower and the *pS129-α-*syn/*α-*syn ratio higher in patients with PD and MSA compared to other study groups. These differences reached statistical significance when patients were grouped based on their presumed underlying proteinopathy (i.e., synucleinopathy vs. tauopathy), with suboptimal specificity. Importantly, when patients with CSF blood contamination were excluded, these differences in *α*-syn and *pS129-α*-syn/*α-*syn ratio were more pronounced, providing improved sensitivity and specificity for a synucleinopathy diagnosis. Still, it has to be noted that *o-α*-syn and *pS129-α*-syn were not different between synucleinopathies and tauopathies, and that the main determining factor of differentiation between the two groups was t-*α*-syn. This index may prove useful, in conjunction with other markers, in the differentiation of PD and MSA on the one hand, and PSP and CBD on the other, a distinction that is not always easy on clinical grounds.

Our findings are in agreement with the literature, with most relevant studies reporting a decrease in *α-*syn and an increase in *pS129-α-*syn and *o-α-*syn in PD [7,8,9]. Despite these statistically significant differences, there is considerable inter-subject overlap in *α-*syn species values among synucleinopathies and other proteinopathies. Thus, at present, CSF *α-*syn species cannot be utilized as clinically relevant biomarkers for the differentiation of synucleinopathies from other disorders, due to their suboptimal diagnostic accuracy. This is partly due to the great variability of α-syn species among patients with PD, as noted elsewhere [15].

Importantly, there were no significant t-*α-*syn alterations in non-synucleinonopathic disorders in our cohort (i.e., AD, FTD, VD, PSP and CBD). This is in general agreement with the literature [7,10,16]. An increase in t-α-syn levels in AD vs. control subjects has been previously reported [19]. Moreover, there are neuropathological data that support the colocalization of *α*-syn and tau oligomers [36], whereas *pS129-α-*syn correlated with Braak stage and Ab levels in another study [37]. A correlation between tau species and *α-*syn has also been reported in AD [19]. In the present study, there were no correlations between *α*-syn species and classical CSF biomarkers in the AD cohort. However, in the absence of a control group, CSF *α-*syn species alterations in non-synucleinopathies could not be further studied in this cohort.

The significant improvement of diagnostic accuracy of *α-*syn and the *pS129-α*-syn/*α*-syn ratio after excluding patients with CSF blood contamination (i.e., >50 RBC) in our cohort, re-emphasizes the need for standardization of pre-analytical factors in CSF biomarker studies [29]. CSF blood contamination in particular is a major confounding factor in α-syn level determination, as has been reported before [38]. The disparity among studies regarding CSF *α*-syn species could in part be attributed to this diversity of peri-analytical factors. These factors may include time of lumbar puncture, type of tubes used for CSF collection, blood contamination of CSF, CSF samples storage and shipment protocol, sample thawing procedure etc. [39]. More importantly, there are differences in analytical factors, such as the type of capture and detection antibodies applied [40].

There have been reports regarding correlations of biomarkers with clinical features. More specifically, it has been postulated that *pS129-α*-syn correlates with clinical severity in PD, as measured by the UPDRS [8]. An increase in *pS129-α-*syn with PD disease progression has also been supported [41]. Moreover, *o-α-*syn correlated with PD motor severity in another study [19]. Phosphorylated and oligomeric α-syn species may vary, depending on disease stage, in PD (i.e., preclinical vs. clinical vs. PD with dementia), raising the question of whether these α-syn species are in fact *state* or *stage* markers [18]. This correlation with disease stage was also evident in asymptomatic LRRK2 mutation carriers, as well as patients with idiopathic REM sleep behavior disorder (iRBD) [42,43,44]. In the present study, CSF *α-*syn species did not correlate with any clinical feature (i.e., disease duration, UPDRS). This is in agreement with a study by Foulds et al., where *α-*syn exhibited stability over PD disease progression [15]. However, only longitudinal studies, with multiple *α-*syn species determinations over different PD disease stages, could provide robust data regarding possible correlations between *α-*syn species and PD severity.

The main focus of this study was the significance of α-syn species as possible biomarkers in neurodegenerative disorders. However, in the cohort of patients with movement disorders, CBD patients had significantly higher τ_T_ values compared to PSP and τ_Τ_/Aβ_42_ ratios compared to PD and PSP. A relative increase in τ_T_ in CBD has been reported previously. The relative increase in τ_T_ in CBD is difficult to interpret, particularly when considering that no significant CSF τ_T_ alteration is evident in PSP, another 4R-tauopathy [30,45].

This study has certain limitations. Initially, the number of patients per study group is suboptimal. This is expected in a single-center cohort, due to the rarity of some of these disorders (e.g., CBD) and due to methodological issues (i.e., CSF blood contamination). Studies of larger cohorts could provide more robust information. Secondly, the absence of a control group precludes the direct comparison of study groups to healthy controls. The inclusion of a control group could further assist in disentangling the complex interplay between CSF biomarkers, clinical phenotype and underlying pathology. Lastly, all diagnoses were based on clinical criteria. To optimize the diagnostic accuracy, all patients fulfilled probable diagnostic criteria and were followed up to exclude patients with atypical clinical features. Moreover, all patients with a non-amnestic phenotype and typical CSF AD profile were excluded, thus minimizing the possibility that the included subjects represented atypical species of AD.

Despite these shortcomings, the present study provides data on CSF *α-*syn species in a variety of neurodegenerative disorders of diverse underlying pathologies and investigates any possible correlations of them with classical CSF AD biomarkers and clinical features. Larger studies with neuropathological data are needed to establish the relationship of CSF biomarkers, clinical phenotype and the underlying pathology.

## 5. Conclusions

Synucleinopathies exhibit lower t-α-syn and higher pS129-α-syn/t-α-syn ratios compared to tauopathies. CSF blood contamination is a major cofounding factor in α-syn species measurements. Non-synucleinopathic disorders do not present with significant CSF α-syn species alterations.

## Figures and Tables

**Figure 1 brainsci-11-00119-f001:**
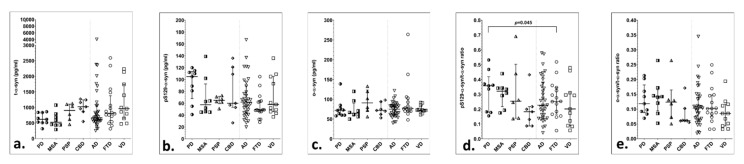
Scatter plots of CSF *α*-syn species and α-syn indices in study groups, with median and 25th–75th interquartile range, after exclusion of patients with CSF blood contamination: (**a**) CSF t-α-syn; (**b**) CSF pS129-α-syn; (**c**) CSF o-α-syn; (**d**) CSF pS129-α-syn to t-α-syn ratio; (**e**) CSF o-α-syn to t-α-syn ratio.

**Figure 2 brainsci-11-00119-f002:**
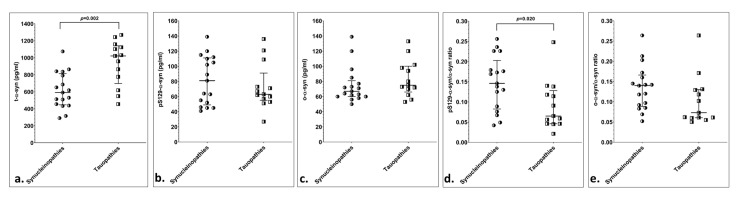
Scatter plots of CSF α-syn species and α-syn indices, with median and 25th–75th interquartile range, based on presumed underlying proteinopathy, after exclusion of patients with CSF blood contamination: (**a**) CSF t-α-syn; (**b**) CSF pS129-α-syn; (**c**) CSF o-α-syn; (**d**) CSF pS129-α-syn to t-α-syn ratio; (**e**) CSF o-α-syn to t-α-syn ratio.

**Table 1 brainsci-11-00119-t001:** Demographic and clinical characteristics of patients. All data are presented as mean (SD) or median (inter-quartile range).

	PD *n* = 13	MSA *n* = 9	PSP *n* = 13	CBD *n* = 9	AD *n* = 51	FTD *n* = 26	VD *n* = 14	*p*-Value
Sex (m/f)	8/5	6/3	9/4	4/5	18/33	15/11	9/5	0.132 †
Age	57.5 (11.1)	61.4 (9.3)	60.8 (5.2)	68.9 (8.0)	66.5 (11.0)	59.6 (8.7)	71.1 (10.9)	0.001 *
Disease duration *(y)*	7.0 (4.0–8.0)	3.0 (1.5–3.0)	3.0 (1.5–3.0)	2.0 (2.0–2.5)	3.0 (2.0–5.0)	3.0 (2.0–5.0)	3.0 (2.0–3.0)	0.010 ‡
UPDRS	24.5 (16.5)	15.4 (17.9)	20.5 (11.4)	26.0 (8.7)	NA	NA	NA	0.442 *
MMSE	28 (26–29)	29 (26–29.5)	24 (22.5–28.5)	24 (19.5–28)	19 (13–23)	24 (17–27)	22 (20–24)	<0.0001 ‡
Clox 2 test	13 (10–13)	13 (12–14.5)	9.5 (8.0–10.0)	8 (2–10)	10 (5–12)	12 (8–14)	8 (7–12)	0.002 ‡
FAB	14.5 (10–16)	14.5 (12.5–16.5)	11 (10–14)	9.5 (6.5–12.5)	10 (7–13)	11 (3–12)	11 (6–12)	0.055 ‡
5-word delayed recall test	5 (5–5)	5 (5–5)	5 (4–5)	4 (3–5)	1.5 (1–3)	4.5 (0–5)	5 (4–5)	0.003 ‡

UPDRS: Unified Parkinson’s Disease Rating Scale; MMSE: Mini Mental State Examination; Clox 2 test: 15-point clock drawing test; FAB: Frontal Assessment Battery; †: x^2^ test; *: analysis of covariance, with sex, age and disease duration as covariates; ‡: independent samples Kruskal–Wallis test.

**Table 2 brainsci-11-00119-t002:** Cerebrospinal fluid biomarkers data of study groups.

	Diagnosis							Analyses		
**I. Classical CSF AD Biomarkers**										
	**PD** ***n* = 13**	**MSA** ***n* = 9**	**PSP** ***n* = 13**	**CBD** ***n* = 9**	**AD** ***n* = 51**	**FTD** ***n* = 26**	**VD** ***n* = 14**	**All Subgroups**	**Movement** **Disorders**	**Dementias**
τ_T_ (pg/mL)	216 (183–259)	305 (211–360)	202 (167–291)	359 (309–454)	649 (477–900)	351 (227–450)	278 (272–347)	<0.0001 *	0.017 *	<0.0001 *
Aβ_42_ (pg/mL)	716 (579–930)	604 (572–819)	690 (559–885)	867 (555–1004)	459 (405–596)	755 (640–896)	651 (570–727)	<0.0001 *	0.891 *	<0.0001 *
τ_P-181_ (pg/mL)	36 (28–49)	37 (47–42)	30 (25–50)	44 (32–104)	72 (57–98)	41 (33–46)	48 (35–56)	<0.0001 *	0.364 *	<0.0001 *
τ_T_/Aβ_42_	0.30 (0.27–0.34)	0.45 (0.32–0.55)	0.33 (0.29–0.40)	0.45 (0.41–0.55)	1.35 (1.00–2.12)	0.47 (0.29–0.71)	0.38 (0.30–0.61)	<0.0001 ‡	0.001 ‡	<0.0001 ‡
τ_P-181_/τ_T_	0.17 (0.15–0.19)	0.12 (0.12–0.16)	0.15 (0.14–0.16)	0.14 (0.13–0.16)	0.12 (0.09–0.14)	0.13 (0.09–0.17)	0.18 (0.13–0.20)	<0.0001 ‡	0.053 ‡	0.022 ‡
**II. CSF α-syn Species in All Patients**										
	**PD** ***n* = 13**	**MSA** ***n* = 9**	**PSP** ***n* = 13**	**CBD** ***n* = 9**	**AD** ***n* = 51**	**FTD** ***n* = 26**	**VD** ***n* = 14**			
t-α-syn (pg/mL)	652 (530–841)	591 (438–686)	882 (618–1102)	958 (549–1158)	690 (574–1240)	792 (539–1063)	908 (600–1564)	0.346 ‡	0.107 ‡	0.653 ‡
pS129-α-syn (pg/mL)	85 (55–110)	54 (46–64)	67 (56–78)	60 (53–109)	59 (47–79)	49 (34–72)	55 (46–93)	0.279 *	0.976 *	0.547 *
ο-α-syn (pg/mL)	75 (64–85)	67 (62–73)	80 (70–111)	72 (62–81)	74 (61–87)	76 (66–88)	73.5 (71–91)	0.753 ‡	0.448 ‡	0.630 ‡
pS129-α-syn/t-α-syn	0.13 (0.05–0.17)	0.14 (0.08–0.18)	0.06 (0.05–0.11)	0.11 (0.05–0.14)	0.09 (0.04–0.14)	0.06 (0.06–0.08)	0.08 (0.03–0.11)	0.190 ‡	0.270 ‡	0.212 ‡
ο-α-syn/t-α-syn	0.11 (0.08–0.14)	0.14 (0.11–0.15)	0.09 (0.06–0.13)	0.06 (0.06–0.10)	0.09 (0.06–0.13)	0.10 (0.08–0.13)	0.09 (0.05–0.13)	0.243 *	0.378 *	0.172 *
**III. CSF α-syn Species after Exclusion of Patients with Blood Contamination**										
	**PD** ***n* = 9**	**MSA** ***n* = 8**	**PSP** ***n* = 6**	**CBD** ***n* = 7**	**AD** ***n* = 37**	**FTD** ***n* = 16**	**VD** ***n* = 11**			
t- a -syn (pg/mL)	618 (513–837)	523 (436–739)	903 (618–1102)	1022 (865–1243)	670 (567–1059)	816 (629–1271)	960 (648–1741)	0.069	0.016	0.275
pS129-α-syn (pg/mL)	105 (81–112)	58 (46–83)	65 (62–71)	60 (53–121)	62 (44–89)	49 (46–64)	58 (46–105)	0.211	0.392	0.368
o-α-syn (pg/mL)	72 (60–77)	65 (59–84)	91 (73–120)	72 (62–94)	75 (61–84)	77 (70–98)	73 (68–85)	0.499	0.379	0.400
pS129-α -syn/t-α-syn	0.17 (0.13–0.23)	0.14 (0.07–0.18)	0.08 (0.06–0.12)	0.06 (0.05–0.14)	0.11 (0.06–0.15)	0.06 (0.04–0.08)	0.09 (0.03–0.11)	0.039	0.130	0.147
o-α-syn/t-α-syn	0.12 (0.09–0.16)	0.14 (0.10–0.16)	0.12 (0.07–0.13)	0.06 (0.06–0.10)	0.10 (0.06–0.14)	0.10 (0.08–0.15)	0.09 (0.04–0.12)	0.223	0.155 *	0.470

All data are presented as median (inter-quartile range). Three analyses were performed: (a) All subgroups: PD vs. MSA vs. PSP vs. CBD vs. AD vs. FTD vs. VD; (b) Movement disorders: PD vs. MSA vs. PSP vs. CBD; (c) Dementias: AD vs. FTD vs. VD; τ_T_: total tau protein; Aβ_42_: amyloid beta with 42 amin acids; τ_P-181_: phosphorylated tau protein at Ser181; t-a-syn: total a-synuclein; pS129-a-syn: phosphorylated a-syn at Ser 129; o-α-syn: a synuclein oligomers; *: analysis of covariance, after log-transformation, with sex, age and disease duration as covariates; ‡: independent samples Kruskal–Wallis test.

**Table 3 brainsci-11-00119-t003:** Cerebrospinal fluid biomarkers data of cohort based on presumed underlying proteinopathy.

**I. Classical CSF AD Biomarkers**			
	**Synucleinopathies** ***n* = 22**	**Tauopathies** ***n* = 22**	***p*-Value**
τ_T_ (pg/mL)	246 (184–309)	278 (192–359)	<0.799 *
Aβ_42_ (pg/mL)	696 (576–895)	774 (557–895)	<0.916 *
τ_P-181_ (pg/mL)	37 (30–46)	43 (28–55)	<0.873 *
τ_T_/Aβ_42_	0.31 (0.29–0.38)	0.39 (0.31–0.45)	<0.175 ‡
τ_P-181_/τ_T_	0.15 (0.12–0.17)	0.14 (0.14–0.16)	<0.918 ‡
t-α-syn (pg/mL)	632 (509–837)	883 (605–1124)	0.034 ‡
pS129-α-syn (pg/mL)	64 (46–105)	65 (53–95)	0.819 *
**II. CSF α-syn Species in All Patients**			
	**Synucleinopathies** ***n* = 22**	**Tauopathies** ***n* = 22**	
o-α-syn (pg/mL)	72.5 (62–85)	76 (62–98)	0.681 ‡
pS129-α-syn/t-α-syn	0.13 (0.07–0.18)	0.08 (0.05–0.12)	0.119 ‡
ο-α-syn/t-α-syn	0.12 (0.08–0.15)	0.08 (0.06–0.13)	0.116 *
**III. CSF α-syn Species after Exclusion of Patients with Blood Contamination**			
	**Synucleinopathies** ***n* = 17**	**Tauopathies** ***n* = 13**	
t-α-syn (pg/mL)	591 (455–791)	1022 (776–1124)	0.002
pS129-α syn (pg/mL)	81 (52–110)	63 (57–73)	0.711
o-α-syn (pg/mL)	67 (60–77)	75 (70–98)	0.229
pS129-α-syn/t-α-syn	0.15 (0.09–0.18)	0.07 (0.05–0.12)	0.020
o-α-syn/t-α-syn	0.14 (0.09–0.16)	0.07 (0.06–0.13)	0.059

All data are presented as median (inter-quartile range). ***τ_T_***: total tau protein; ***Aβ_42_***: amyloid beta with 42 amin acids; ***τ_P-181_***: phosphorylated tau protein at Ser181; ***a-syn***: total a-synuclein; ***pS129-a-syn***: phosphorylated a-syn at Ser 129; ***ο-α-syn***: a synuclein oligomers with syn-O_2_ as capture antibody; *: analysis of covariance, after log-transformation, with sex, age and disease duration as covariates; ‡: independent samples Kruskal-Wallis test.

## Data Availability

The data presented in this study are available on request from the corresponding author. The data are not publicly available due to privacy restrictions.

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
