# Peer review of "Cerebrospinal Fluid α-Synuclein Species in Cognitive and Movements Disorders"

_brainsci, 2021, doi:10.3390/brainsci11010119_

Round 1
Reviewer 1 Report
The question of whether and how α-synuclein conformers in CSF can be used as biomarkers for various proteinopathies is an important one. The present study provides useful information in this regard, yet stands to be significantly improved with the inclusion of a control group, reorganization of the tables/figures, and additional discussion and clarifications.
Major comments:
While the comparisons between different disease groups is valuable, it is severely undercut by the lack of inclusion of a healthy, age-matched control group in this study. Without a control group, it is difficult to put the comparisons between disease groups into any context. Are the observed differences meaningful when a control group is included?
It is not clear why the authors decided to include blood contaminated-samples in the main figures while putting all of the data excluding blood contaminated-samples into the supplement. The supplementary tables and figures are arguably much more meaningful in that they show a greater number of significant differences in α-synuclein conformers, as well as differences that are more highly significant than those reported in the main figures. I suggest switching these data, or including all of the data in the main text.
Did the authors attempt any comparisons using non-classical, potentially unexpected ratios such as p-S129-α-syn/ o-α-syn, or o-α-syn/ tau? While some ratios may seem unlikely as possible biomarkers, the discovery of any such unexpected patterns would be highly novel and exciting.
The potential use of tau as a biomarker in movement disorders is intriguing (Table 2). Please comment.
It is unclear why the authors do not report tau or Aβ values in Sup. Table 2.
It is not clear how the authors segregated patients into “synucleinopathy” versus “tauopathy.” It is described as both “known” as well as “presumed” which seem contradictory. Did the authors confirm the type of proteinopathy, and if so, how? Or was it presumed based on diagnosis?
Minor comments:
Consider using the word “conformers” or “species” to describe α-synuclein rather than “forms”. The phrase “α-synuclein forms” is quite vague.
Please indicate statistical significance directly on the scatterplots in order to improve clarity.
In line 59, I think “α-syn o-α-syn” should simply be “o-α-syn”?
Should the end of line 171 read “e)” instead of “s)”?
In line 233, I think “o-α-syn o-“ should simply be “o-α-syn”?
Author Response
Response to Reviewers
Reviewer 1
Comment 1: “While the comparisons between different disease groups is valuable, it is severely undercut by the lack of inclusion of a healthy, age-matched control group in this study. Without a control group, it is difficult to put the comparisons between disease groups into any context. Are the observed differences meaningful when a control group is included?”
Response:
Indeed, the absence of an age-matched control group is a significant limitation of this study, as analyzed in the Discussion section: “Secondly, the absence of a control group precludes the direct comparison of study groups to healthy controls. The inclusion of a control group could further assist in disentangling the complex interplay between CSF biomarkers, clinical phenotype and underlying pathology”.
Despite this limitation, we believe that the direct comparison among different disease groups allows for certain conclusions regarding α-syn species, as analyzed in the Discussion section.
Comment 2: “It is not clear why the authors decided to include blood contaminated-samples in the main figures while putting all of the data excluding blood contaminated-samples into the supplement. The supplementary tables and figures are arguably much more meaningful in that they show a greater number of significant differences in α-synuclein conformers, as well as differences that are more highly significant than those reported in the main figures. I suggest switching these data or including all of the data in the main text”.
Response:
Thank you for your comment. In accordance with your comment, all CSF data of the cohort after excluding patients with CSF blood contamination were included in Tables 2 and 3. Furthermore, tables 2 and 3 were subdivided in three sections for clarity:
- Classical CSF AD biomarkers
- CSF α-syn species in all patients
- CSF α-syn species after exclusion of patients with blood contamination
Supplementary Tables 2 and 3 were deleted, since this data was incorporated in Tables 2 and 3 respectively.
Figures 1 and 2 were modified to include exclusively patients with no blood contamination, and supplementary Figures 1 and 2 were deleted.
Comment 3: “Did the authors attempt any comparisons using non-classical, potentially unexpected ratios such as p-S129-α-syn/ o-α-syn, or o-α-syn/ tau? While some ratios may seem unlikely as possible biomarkers, the discovery of any such unexpected patterns would be highly novel and exciting”.
Response:
All possible combinations of biomarkers were tested, including the p-S129-α-syn/ o-α-syn, or o-α-syn/ tau ratios. None of these ratios provided significant differences. Only ratios with a plausible biological basis were included in the Tables for brevity and clarity. If it is considered necessary, more ratios could be included.
Comment 4: “The potential use of tau as a biomarker in movement disorders is intriguing (Table 2). Please comment”.
Response:
Thank you for your insightful comment. Indeed, tau protein may be useful in differentiating movement disorders. We have previously noted a relative increase in τT in MSA compared to PD, and the potential of the τT/Aβ42 ratio as a biomarker for the differentiation of MSA-P from PD (Constantinides VC, Paraskevas GP, Emmanouilidou E, Petropoulou O, Bougea A, Vekrellis K, et al. CSF biomarkers beta-amyloid, tau proteins and a-synuclein in the differential diagnosis of Parkinson-plus syndromes. J Neurol Sci. 2017;382:91-5). Moreover, τT has been reported to increase in CBS patients, although these studies may have included Alzheimer’s disease patients with a CBS phenotype.
The following paragraph was added in the Discussion section:
“The main focus of this study was the significance of α-syn species as possible biomarkers in neurodegenerative disorders. However, in the cohort of patients with movement disorders, CBD patients had significantly higher τT values compared to PSP and τΤ/Aβ42 ratios compared to PD and PSP. A relative increase of τT in CBD has been reported previously. The relative increase of τT in CBD is difficult to interpret, particularly when considering that no significant CSF τT alteration is evident in PSP, another 4R-tauopathy[30, 45]”.
The following reference was added:
Urakami, K., et al., A comparison of tau protein in cerebrospinal fluid between corticobasal degeneration and progressive supranuclear palsy. Neurosci Lett, 1999. 259(2): p. 127-129.
Comment 5: “It is unclear why the authors do not report tau or Aβ values in Sup. Table 2”.
Response:
CSF blood contamination is a major cofounding factor in α-syn species measurement [Barbour, R., et al., Red blood cells are the major source of alpha-synuclein in blood. Neurodegener Dis, 2008. 5(2): p. 55-9.].
CSF blood contamination does not affect CSF tau or Aβ levels. For this reason, repeating classical CSF AD biomarkers measurements after exclusion of patients with CSF blood contamination would not add any further information to the initial analysis.
Comment 6: “It is not clear how the authors segregated patients into “synucleinopathy” versus “tauopathy.” It is described as both “known” as well as “presumed” which seem contradictory. Did the authors confirm the type of proteinopathy, and if so, how? Or was it presumed based on diagnosis?”
Response: Patients were segregated into synucleinopathy or tauopathy based on established clinical diagnostic criteria. All patients fulfilled criteria for probable PSP, CBD, MSA and clinically established PD. Only patients with Richardson’s syndrome were included in the case of PSP. The specificity and positive predictive value of these criteria is exceptionally high, rendering the possibility of a misdiagnosis unlikely. Moreover, all patients were followed-up in order to exclude any patients with atypical clinical features.
The problem of an accurate in vivo diagnosis is particularly problematic in CBS patients. Pathological studies have concluded that a CBS patient may harbor a CBD, PSP or AD diagnosis. Other underlying pathologies are exceedingly rare (e.g. FTD-TDP-43). All CBS patients in our cohort with a typical CSF AD biomarker profile were excluded from analyses, as described in detail in the Patients subsection of the Materials and Methods section. Thus, it is highly probable that a patient with a CBS phenotype had an underlying 4-R tauopathy.
Some relevant references on the issue:
- O'Sullivan SS, Massey LA, Williams DR, et al. Clinical outcomes of progressive supranuclear palsy and multiple system atrophy. Brain: 2008;131(Pt 5):1362-1372.
- Respondek G, Roeber S, Kretzschmar H, et al. Accuracy of the National Institute for Neurological Disorders and Stroke/Society for Progressive Supranuclear Palsy and neuroprotection and natural history in Parkinson plus syndromes criteria for the diagnosis of progressive supranuclear palsy. Movement disorders: 2013;28(4):504-509.
- Ling H, O'Sullivan SS, Holton JL, et al. Does corticobasal degeneration exist? A clinicopathological re-evaluation. Brain: 2010;133(Pt 7):2045-2057.
- Wenning GK, Ben Shlomo Y, Magalhaes M, Daniel SE, Quinn NP. Clinical features and natural history of multiple system atrophy. An analysis of 100 cases. Brain: 1994;117 ( Pt 4):835-845.
- Osaki Y, Ben-Shlomo Y, Lees AJ, Wenning GK, Quinn NP. A validation exercise on the new consensus criteria for multiple system atrophy. Movement disorders: 2009;24(15):2272-2276.
In accordance with your comment, we have used the term “presumed” proteinopathy throughout the manuscript.
The following paragraph in the Patients subsection of the Materials and Methods Section was altered from:
“Patients fulfilling established probable criteria for PD[18], MSA[19], AD[5], progressive supranuclear palsy (PSP)[20], corticobasal degeneration (CBD)[21], frontotemporal dementia (FTD)[22] and vascular dementia (VD)[23] were included”.
to:
“Patients fulfilling established clinical diagnostic criteria for clinically established PD[18], and probable MSA[19], AD[5], Richardson’s syndrome of progressive supranuclear palsy (PSP)[20], corticobasal degeneration (CBD)[21], frontotemporal dementia (FTD)[22] and vascular dementia (VD)[23] were included”.
Comment 7: “Consider using the word “conformers” or “species” to describe α-synuclein rather than “forms”. The phrase “α-synuclein forms” is quite vague”.
Response: The word “species” was used throughout the manuscript to substitute the word “forms”.
Comment 8: “Please indicate statistical significance directly on the scatterplots in order to improve clarity”.
Response:
Statistical significance was added directly on the scatterplots, as suggested.
Comment 9: “In line 59, I think “α-syn o-α-syn” should simply be “o-α-syn”?”
Response:
This error was corrected
Comment 10: “Should the end of line 171 read “e)” instead of “s)”?”
Response:
This error was corrected
Comment 11: “In line 233, I think “o-α-syn o-“ should simply be “o-α-syn”?”
Response:
This error was corrected.
Reviewer 2 Report
The study is very interesting and well designed. The methodology used is correct, and the manuscript is properly organized and written. For this reason, this reviewer considers that the manuscript, in the present form, is acceptable for publication in Brain Sciences, after the appropriate editorial revision of minor grammar or spell mistakes.
Author Response
Response to Reviewers
Reviewer 2
“The study is very interesting and well designed. The methodology used is correct, and the manuscript is properly organized and written. For this reason, this reviewer considers that the manuscript, in the present form, is acceptable for publication in Brain Sciences, after the appropriate editorial revision of minor grammar or spell mistakes”.
Response: Thank you for your kind remarks.
Reviewer 3 Report
The authors present a manuscript describing the quantification of different forms of a-syn and tau in the csf of neurological patients comparing different diseases with different underlying pathologies (PD, MSA, PSP, CBD, AD, FTD, VD). Even though the significant differences presented here are limited, the study was performed soundly and the patient cohorts and methods were described well. The limitations of the study, of which the very low number of participants is the most severe one, are properly discussed.
The presentation of the data is well-arranged, however the authors should consider to include suppl. fig.2 into the main manuscript, as the exclusion of blood contamination is very important.
Generally the graphs are too small and particularly the axis labels need to be increased.
The authors need to improve the mode of citation and provide reference to original papers instead of reviews (reference 7). Particularly in the discussion section (but also in the introduction), the authors need to discuss the different results individually refering to the original data instead of just combining all results and citing a review.
Author Response
Response to Reviewers
Reviewer 3
Comment 1: “The presentation of the data is well-arranged, however the authors should consider to include suppl. fig.2 into the main manuscript, as the exclusion of blood contamination is very important”.
Response:
Supplementary Tables 2 and 3, which include data of the cohort after exclusion of patients with CSF blood contamination have been incorporated to Tables 2 and 3 (please refer to response to Comment 2 of Reviewer #1 for a more detailed response). Figures 1 and 2 were modified to include data on α-syn indices. Moreover, Figures 1 and 2 were modified to include exclusively patients with no blood contamination.
Comment 2: “Generally the graphs are too small and particularly the axis labels need to be increased”.
Response:
All Figures were modified to a 1200 dpi TIFF format. These Figures can be expanded significantly, based on the Editor’s decision. All axis labels’ fonts were increased to 24 from 16 for improved clarity. All points in the scatterplots were enlarged.
Comment 3: “The authors need to improve the mode of citation and provide reference to original papers instead of reviews (reference 7). Particularly in the discussion section (but also in the introduction), the authors need to discuss the different results individually refering to the original data instead of just combining all results and citing a review”.
Response:
We have added some of the original studies regarding α-syn species, both in the Introduction and the Discussion section. The following references were added:
- Fujiwara, H., et al., alpha-Synuclein is phosphorylated in synucleinopathy lesions. Nat Cell Biol, 2002. 4(2): p. 160-4.
- Nishie, M., et al., Accumulation of phosphorylated alpha-synuclein in the brain and peripheral ganglia of patients with multiple system atrophy. Acta Neuropathol, 2004. 107(4): p. 292-8.
- Kazantsev, A.G. and A.M. Kolchinsky, Central role of alpha-synuclein oligomers in neurodegeneration in Parkinson disease. Arch Neurol, 2008. 65(12): p. 1577-81.
- Tokuda, T., et al., Detection of elevated levels of alpha-synuclein oligomers in CSF from patients with Parkinson disease. Neurology, 2010. 75(20): p. 1766-72.
Round 2
Reviewer 1 Report
The authors have made several improvements to the organization and presentation of their data, as well as the clarity of the text. In particular, the new versions of their main tables show the data clearly and comprehensively, with subsections for classical AD markers, asyn species in all samples, and asyn species only in samples with no blood contamination.
All of my concerns have been adequately addressed. I recommend the paper for publication.